# Chemistry of Polythiols and Their Industrial Applications

**DOI:** 10.3390/ma17061343

**Published:** 2024-03-14

**Authors:** Seung-Mo Hong, Oh Young Kim, Seok-Ho Hwang

**Affiliations:** 1Department of Polymer Science & Engineering, Dankook University, Yongin 16890, Republic of Korea; ssendear@naver.com (S.-M.H.); koybon@dankook.ac.kr (O.Y.K.); 2Advanced Materials Division, Shin-A T&C Co., Ltd., Seoul 08501, Republic of Korea

**Keywords:** polythiol, thiol-ene, thiol-isocyanate, thiol-epoxy, industrial applications

## Abstract

Thiols can react with readily available organic substrates under benign conditions, making them suitable for use in chemical, biological, physical, and materials and engineering research areas. In particular, the highly efficient thiol-based click reaction includes the reaction of radicals with electron-rich enes, Michael addition with electron-poor enes, carbonyl addition with isocyanate S_N_2 ring opening with epoxies, and S_N_2 nucleophilic substitution with halogens. This mini review provides insights into emerging venues for their industrial applications, especially for the applications of thiol-ene, thiol–isocyanate, and thiol–epoxy reactions, highlighting a brief chemistry of thiols as well as various approaches to polythiol synthesis.

## 1. History of Sulfur Compounds

The sulfur element is abundant on earth, and it has been mined for thousands of years. Until the early 1900s, the source of sulfur was mainly the soil surrounding the volcano on the Mediterranean island of Sicily [1]. Owing to its unique properties, sulfur has been used as a raw material for numerous products from the weapons and tools of the Bronze Age to 18th-century gunpowder and modern sulfuric acid [2].

In the early-to-mid 20th century, numerous studies focused on understanding the phases of sulfur [3,4]. Sulfur exists as a yellow orthorhombic crystalline solid at temperatures below 96 °C [5]. Sulfur remains yellow between 96 °C and 119 °C and exists in the form of a monoclinic crystalline solid, and both orthorhombic and monoclinic sulfur phases exhibit the S8 ring structure [6]. Sulfur begins to melt into a yellow liquid at approximately 159 °C when the S8 ring begins to open, forming an amorphous and viscous sulfur chain [6,7]. As it is heated further, its color darkens from yellow to orange and eventually turns red [7]. Usually, other elements evaporate as the temperature rises, but the viscous sulfur is polymerized at 200 °C into a red solid substance [6]. Upon cooling, this change is reversed, and the amorphous red polymer returns to the orthorhombic crystalline yellow solid at low temperatures [7,8]. Meyer reported that when quenched with a thin film at 200 °C, it became yellow rather than red [9,10]. Meyer concluded that the overall red color was due to the presence of organic impurities and small sulfur molecules, such as S3 and S4 [9,10]. The color change of sulfur has remained a subject of controversy.

Sulfur is stable despite its temperature-dependent phase change, but it also forms compounds with numerous other elements under suitable conditions, including galena (PbS), gypsum (CaSO_4_·2H_2_O), pyrite (FeS_2_), and hydrogen sulfide (H_2_S). This ability to combine with other elements makes sulfur behave in various ways but hinders its extraction at the element level [8]. In fact, after an 18th-century chemist, Antoine Lavoisier, started studying sulfur, it was no longer considered a compound but was finally recognized as an element [11]. Since then, various methods have been developed to utilize sulfur, and various compounds involving sulfur have been developed, including iron–sulfur clusters [12,13], molybdenum disulfide [14,15,16], thiophene-based conductive polymers [17,18], cysteine, and methionine.

Recently, sulfur has been recovered as a major byproduct of the refining and natural gas purification processes. It is estimated that over 60 million tons of sulfur are produced annually [19]. In addition to ongoing efforts to utilize the unique properties of sulfur, extensive research in polymer chemistry and materials engineering is devoted to developing energy-efficient and environmentally friendly approaches to utilize the recovered sulfur element.

## 2. Comparison of the Physical Properties of Thiols and Alcohols

Thiols are analogues of alcohols in which the oxygen atom is replaced by a sulfur atom, and traditionally, it is also called mercaptan. And, organosulfur compounds in which two or more thiol functional groups are attached to a carbon atom of any aliphatic or aromatic moiety are defined as polythiols. Polythiols can react with various functional groups to produce a thermosetting polymer.

Methanol (CH_3_OH) has a boiling point of 64 °C, whereas methanethiol (CH_3_SH) has a boiling point of 5.8 °C, and ethanol (CH_3_CH_2_OH) has a boiling point of 78 °C, whereas ethanethiol (CH_3_CH_2_SH) has a boiling point of 37 °C, indicating that although the oxygen and sulfur atoms are in the same group (Group 16) of the periodic table, the bonding of sulfur compounds via hydrogen bonding is not as extensive as that of alcohols. This is because the sulfur atom has a larger size and lower electronegativity than the oxygen atom, making the electrostatic force between the sulfur atom and the hydrogen atom relatively small [20]. Thus, a large difference exists between the reactivity of an alcohol and a thiol, and their applications vastly differ as well.

The bond strength of S-H bond is much less than the O-H bond as reflected in their respective bond dissociation energies. This means that the free-radical hydrogen abstraction for S–H is easier compared to that for O–H. For example, the bond dissociation energy of CH_3_S-H is 89 kcal/mol, while that of CH_3_O-H is 100 kcal/mol [21,22]. Generally, the geometry of the thiol group is much closer to a linear 90° angle than that of a hydroxyl group (105° angle) [23,24], because the sulfur atom has much more *p*-character comparing with that of the oxygen atom. The alcohol containing hydroxy group has strong intermolecular force through a hydrogen bonding, but the intermolecular force of the thiols is much weaker than those in alcohol counterparts [25,26]. For example, the equilibrium constant for the equilibrium state between monomeric and dimeric forms of thiophenol is only 0.0001 mol^−1^ in CCl_4_ at 25 °C. Also, the weaker bonding strength of S-H in comparison with that of O-H makes thiols more acidic than corresponding alcohols [27,28,29,30,31,32]. Interestingly, the difference between the *p*Ka values of butanol and phenol is greater (*∆p*Ka≃6) than that between the *p*Ka values of their corresponding sulfur analogs (*∆p*Ka≃4). This phenomenon can be explained by the greater contribution of the canonical form (b) and (c), which is showing the greater contribution at the oxygen compound over its sulfur analog. (Figure 1) The alcoholate anion is a much stronger base than the corresponding sulfur analog, and the former has greater affinity toward a proton than the latter. However, the order of nucleophilicities or anions is reversed [33].

## 3. Synthesis of Thiol Compounds

Since many review papers on the synthesis of thiol compounds have been published [34,35,36], in this review, the synthesis methods of thiol compounds were briefly investigated by reaction type.

### 3.1. Nucleophilic Substitution 

Thiols are directly obtained by the reaction of organic halides with metallic hydrosulfides [37]. (Figure 2) Alkyl sulfates react similarly. Alkyl sulfides are often the by-product of this reaction [38,39,40,41].

The reaction of alkyl halide with thiourea followed by the alkaline hydrolysis of the intermediate isothiouronium salt is one of the most common methods of thiol preparation [42,43] (Figure 3).

The ring-opening reaction of a heterocyclic three-member ring may be classified in these categories [34,44,45,46] (Figure 4).

### 3.2. Addition to Olefins

The addition of hydrogen sulfide to an olefin affords a small yield of thiol, which has the structure predicated by Markovnikov’s rule [47,48] (Figure 5).

A more sophisticated method in this category is the radical addition of thioacetic acid followed by hydrolysis [49], although this reaction results in an anti-Markovnikov adduct (Figure 6).

### 3.3. Reduction Reaction

The reduction of sulfonyl chloride is used for synthesizing aromatic thiols [50,51]. The reducing agents may be Zn–HCl, LiAlH, and similar compounds. Disulfides can also be used as precursors for the synthesis of thiols [52] (Figure 7).

### 3.4. From Organometallics

Thiols are synthesized by the reaction of the elemental sulfur with the Grignard reagent [53]. Organolithium compounds work similarly [54,55,56,57] (Figure 8).

### 3.5. Other Methods

Several other methods can be used for synthesizing thiols, especially those in which a functional group is separated from a sulfur atom. Thus, thiocyanates [52,58] and thiosulfates [59,60] yield thiols (Figure 9).

## 4. Reaction of Thiol Compounds

Thiols are soft nucleophiles compared to alcohols and amines [61]. Since the thiolate anion and the thiyl radical have very high reactivity, they show high yield and fast reactivity under relatively mild conditions, and the resulting thiol reaction can be classified as a “click” reaction. The chemical structure of a thiol greatly influences its physicochemical properties and reactivity; it can strongly change not only the electrophilicity of the thiyl radical but also the dissociation energy of the S–H bond [62,63,64]. The *p*Ka values of thiols are strongly influenced by the chemical structure, which affects the nucleophilicity of the thiol and thiolate.

Thiols react via two mechanisms: radicals and nucleophiles [64]. A thiol radical addition includes a thiyl radical and can be initiated by heat or light [63,65,66,67]. The most important radical reaction is the thiol–ene addition reaction, which satisfies the criteria proposed by Sharpless et al. in 2001 and can be classified as a click reaction [68]. Another radical reaction is a thiol–yne addition reaction, allowing two thiols to bind to a triple bond [69,70].

A nucleophilic reaction involves a thiolate anion and a thiol [64]. When a basic compound is present as a catalyst, the thiol is generally deprotonated to give a reactive thiolate. The thiol–nucleophilic addition reaction [71] includes the thiol–epoxy nucleophilic ring opening reaction [72], thiol–isocyanate reaction [73,74], and thiol–Michael addition reaction [75,76,77,78,79] (Figure 1). The thiol–Michael addition reaction is proceeded with (meth)acrylate, acryl amide, maleimide, maleate, acrylonitrile, vinyl sulfone, and oxazoline [71,76].

## 5. Polythiol Compounds

### 5.1. Thioether Polythiol Compounds

It is possible to synthesize a thioether polythiol, a polythiol compound containing a sulfide bond, for various purposes. A thioether polythiol containing three or more thiol functional groups can be prepared using a general polythiol synthesis method [80,81]. Normally, an alcohol- or halogen-containing sulfide reacts with thiourea to synthesize an isothiouronium salt. The thioether polythiol is acquired by the hydrolysis of this isothiouronium salt [82]. The general thioether polythiol synthesis method is shown in Figure 10 [83,84,85,86].

The synthesis of a thioether polythiol compound by producing an isothiouronium salt from an organic halide or alcohol offers high yield, producing a small amount of by-product, and excellent operability, resulting in higher product quality compared to other production methods. Therefore, this method is one of the best methods used to produce a thioether polythiol compound.

### 5.2. Ester Polythiol

An ester polythiol compound can be synthesized using a direct esterification of a mercaptan compound containing a carboxylic acid and a polyalcohol compound at relatively low temperature, and this reaction is a reversible reaction in which water is produced as a by-product. Compounds containing a mercaptan compound and a carboxylic acid are called mercapturic acid, such as thioglycolic acid, 3-mercaptopropionic acid, and 3-mercaptobutanoic acid.

Unlike their carboxylic acid analogs, for mercapturic acids, the reaction temperature must be minimized to avoid the unwanted reaction of thiols, such as the formation of disulfides. This means that catalysts must be used at low temperatures utilized for ester polythiol production rather than higher temperatures commonly employed in the esterification reaction. An efficient, high-temperature (160–240 °C) esterification catalyst may have inferior low-temperature performance compared to a poorer high-temperature catalyst, which is more efficient at the low temperatures used to manufacture ester polythiols. Thus, no direct correlation of catalytic effectiveness exists between the processing conditions normally used and those used in ester polythiol preparation. The unaided removal of water at atmospheric pressure cannot be achieved at temperatures below 115–120 °C, and a significantly higher temperature must be used, usually combined with an inert carrier gas to carry the water over. Even in these high-temperature preparations, solvents are frequently used to assist in removing water by distillation. An azeotrope may be defined as a water–solvent mixture, which boils at a temperature below the boiling point of either the solvent or water, such that water and the solvent cannot be separated by distillation alone. An example of an azeotrope is 4% water–96% ethanol. Thus, distilling a solvent that forms an azeotrope with water carries water with it in the vapor phase, thereby removing it from the reaction system to drive the reaction toward completion. Azeotropic distillation is essential for the removal of water at reaction temperatures below or around 100 °C.

Although the carboxylic acid–hydroxyl reaction to form an ester is, perhaps, one of the best documented and most well-known organic chemistry reactions, its utilization for ester polythiol formation is restricted by numerous side reactions, as in transesterification. Kemmerer et al [87]. claim that direct esterification can be displayed at low temperatures.

The traditionally accepted mechanism of the carboxylic acid–hydroxyl reaction is displayed in Figure 10 [88]. Although the reaction resembles an S_N_2 displacement, it is an addition elimination reaction rather than an inversion process [88]. Both esterification and hydrolysis reactions are very slow in the absence of a catalyst because the acid catalyst can protonate the unshared electron pair of the oxygen in both carboxylic acids and alcohols. Although the activity of the latter is reduced toward the carbonyl atom of the acid at a low acid concentration of the catalyst, the protonation of the carbonyl oxygen of the carbonylic acid, shown in Figure 11a, greatly enhances the electron-pair-accepting power of the carbonyl carbon. Intermediate Figure 11b rapidly and reversibly forms intermediate Figure 11c upon the simple proton transfer shown in Figure 11.

The esterification reaction is an equilibrium process, and an excess of one reactant must be used to favor ester formation. The removal of water in esterification shifts the equilibrium and thus drives the reaction toward ester formation. The alcohol used for esterification reactions must not undergo extensive side reactions in strong acids. This ester formation mechanism is susceptible to steric hinderance [88], because intermediates (b) and (c) shown in Figure 11 cause instability owing to the crowding of the substituent group. Thus, not surprisingly, the order of reactivity of hydroxyl groups in alcohols is Primary > Secondary > Tertiary. Figure 2 shows the commercial grades of ester polythiol compounds.

## 6. Applications of Polythiol Compounds

### 6.1. Ultraviolet Curing Area

Radical-based photopolymerization including acrylate and methacrylate can provide both spatial and temporal control through a chain-growth mechanism, and this reaction is used in various applications, such as coatings, dental materials, contact lenses, and photolithographic processes [89,90,91]. Recently, the applications of photopolymerization have been expanded to nanotechnology, biomaterials, high-resolution lithography, selective functionalization of preformed linear polymers, surface functionalization, optics, low-stress networks, and high-impact energy-absorbing thermosetting resins. However, the conventional photopolymerization reaction has limitations, including oxygen inhibition [92,93], shape deformation due to curing shrinkage, stress generation [94,95], complex polymerization kinetics [96,97,98], and so on [99,100,101].

The general concept of the reaction between a carbon–carbon double bond and a thiol, that is, thiol–ene, was well known in the early 1900s [102]. In Figure 12, the thiol–ene reaction can be explained by two reactions: one is a thiol–ene free-radical addition to an electron-rich carbon–carbon double bond, and the other is the carbon–carbon double bond of the electron-poor catalyzed thiol–Michael addition reaction.

The thiol–ene reaction proceeded by radicals (thiol–ene reaction) or anionic chains (thiol–Michael addition reaction) has the characteristics of a click reaction. This reaction proceeds insensitively to the surrounding oxygen and water. In particular, an effective reaction yield can be obtained for acrylate-based ultraviolet (UV) radical curing, which is affected by the curing yield due to oxygen inhibition [103]. The biggest advantage of the thiol–ene reaction is that thiol and ene compounds can be easily obtained commercially because the synthesis of new multipurpose thiols and enes is relatively easy, and new monomer systems are being developed widely. For example, an ally-ether-terminated biodegradable polyester was used as the ene component of a thiol–ene photocurable system [104] where a novel low-molecular-weight thiol was also used for the synthesis of linear polysulfides [105].

UV curing is an effective crosslinking method used in various applications, such as coatings, adhesives, printing inks, plastic lenses, optical fibers, dental materials, and wood composites [106,107,108,109,110]. The UV curing method is generally based on the free-radical polymerization of polyfunctional ene monomers and offers shorter reaction time, lower energy consumption, and lack of reliance on volatile organic solvents compared to conventional thermal curing and solvent-based lacquer methods. Although the UV curing method has been widely used, it also has limitations. Among them, oxygen inhibition is the most important obstacle of the UV curing method (Figure 13). Oxygen forms peroxide radicals in the initiation and growth reactions in the free-radical polymerization reaction and lowers the curing density by lowering the reactivity of the free-radical polymerization [91,111,112]. Currently, the best way to solve this problem is to proceed with UV curing in an inert gas (nitrogen, argon, etc.) atmosphere, or use a large amount of a photoinitiator. In addition, it results in significant volumetric shrinkage and internal stress because gel points reflect a transition generally observed at low cure yields [63]. However, the photopolymerization of polythiol and ene (thiol–ene click reaction) is an effective method for rapidly producing films and cured products with excellent physical and mechanical properties, as shown in Figure 14 [63,64,113,114,115,116,117,118]. As ene and thiol react stepwise via an exceedingly fast chain transfer reaction, the curing mechanism differs from the general UV curing mechanism [103,119,120]. This reaction includes adding a thiyl radical to the carbon of the ene functional group, generating a thiyl radical due to hydrogen abstraction to the thiol group by a radical of a carbon–carbon double bond. The growth and chain transfer reactions can be explained using the basic thiol–ene polymerization reaction mechanism, and finally, the termination reaction occurs owing to the radical–radical coupling reaction. The thiol–ene click reaction shows similar reaction rates in inert gas and air. Since the thiol group reacts with the peroxide radical and forms a highly reactive thiyl radical after hydrogen abstraction due to the peroxide radical, the oxygen inhibition problem can be addressed [111,121]. In addition, direct UV irradiation can induce a reaction without a photoinitiator [122]. The thiol–ene system can exhibit an intrinsically fast reaction rate [119], low polymerization shrinkage [114], and the formation of a uniform polymer structure. Therefore, thiol–ene photopolymerization has recently been commercialized and widely studied in many fields, such as clear coatings, inks, adhesives, negative photoresists, and optical materials [123,124]. Almost all types of ene compounds, including acrylates, vinyl ethers, allyl ethers, vinyl esters, and alkenes, can react efficiently with polythiols [66,98,125]. As the effect of polythiol and ene structure on the overall rate of the thiol–ene addition reaction have been studied extensively, it is possible to select various materials with a combination of characteristics necessary for a specific application [63,126].

The reactivity decreases as the electron density of the carbon–carbon double bond decreases. In the case of reacting with a thiyl radical with an extremely conjugated carbon–carbon double bond (conjugated diene), it is known that copolymerization with thiol can be performed very slowly because of the stability of the radical formed. As a result, the reaction sequence of acrylate and allyl-ether-type ene to polythiol significantly differ [127], and competitive homopolymerization may occur in acrylate systems [128].

### 6.2. High Refractive Index Plastic Spectacle Lenses for Vision Correction

Optically transparent polymers have been developed as a lens material to replace inorganic glass. Owing to their excellent lightness, impact resistance, processability, and dyeability [84,129,130], these materials are being utilized in various fields such as optical adhesives, optical filters, encapsulants for light-emitting diodes (LEDs), prisms, light guide plates, and antireflection coatings [131]. Although many studies are being conducted to increase the refractive indices of transparent polymers, the traditional polymers exhibit low refractive indices. For example, the refractive indices of polymethyl methacrylate (PMMA), polycarbonate (PC), and polystyrene (PS) are 1.49, 1.58, and 1.59, respectively [132].

A polymer with a high refractive index can be prepared via molecular design and synthesis based on the Lorentz–Lorenz equation. This method is used to design high refractive index polymeric materials with an aromatic ring [133], a halogen atom excluding fluorine [134], a sulfur atom [135], and a phosphorus atom [136], and introducing an atom or substituent with a small molar volume result in a high refractive index. Representative high refractive index polymers are polyimide (PI), poly(phenylene thioether), and poly(thioether sulfone), which involve thioether, sulfone, and triazine units, respectively, and can be synthesized by introducing a rigid group [137,138,139,140,141].

In general, typical polymers that have good impact resistance and light and optical transparency include PMMA, PC, etc. In the early days of plastic lenses, aryl diglycol carbonate (ADC) resin developed by PPG Industries Inc. was commercialized under the product name “CR-39”.

Compared to glass lenses, plastic lenses exhibit a low specific gravity of 1/2 and a high refractive index, so their thickness can be reduced, and they provide good impact strength and are not easily broken [142].

The Abbe number is a measure of light dispersion and is very important for optical lenses used in the visible light region. As the Abbe number increases, the dispersion of light decreases. In general, as the refractive index increases, the Abbe number tends to decrease [143]. It is important to balance the Abbe number and the refractive index in optical polymers, which is enabled by using sulfur to increase the refractive index without sacrificing the Abbe number. Thus, research on high refractive index polymers containing sulfur has shown remarkable progress recently. Among them, polythiourethane containing sulfur [82,83,84,85] has been produced as a highly refractive lens for vision correction owing to its abrasion resistance, impact resistance, transparency, high refractive index, and simple production process [144,145].

The click reaction proposed by Sharpless et al [68] is an efficient and diverse modular reaction based on stereospecificity, which has drawn immense interest in biopolymer synthesis and polymer chemistry [146,147,148]. A thermosetting polythiourethane formed by the click reaction of polythiol and isocyanate monomers [149,150] was first commercialized by Mitsui Chemicals in the late 1980s owing to its excellent impact resistance, high refractive index, and simple production process. The polythiourethane lens developed by Mitsui Chemicals, under the trade name ‘MR series’ is a plastic lens with a high refractive index, transparency, and Abbe number. The composition of this poly(thiourethane) plastic lens and its optical properties are presented in Table 1. Here, when a sulfur atom with a high molar refractive index and a benzene ring structure with a high π electron density are simultaneously introduced, the refractive index of polymer can be increased without sacrificing the Abbe number. This can be compared with plastic lenses, such as polycarbonate (refractive index: 1.59, Abbe number: 30), highly refractive acrylic lens (refractive index: 1.60, Abbe number: 32), and ADC (refractive index: 1.50, Abbe number: 57) [151].

Polythiourethane lenses involve aromatic or aliphatic isocyanates, thioether polythiols, and ester polythiols individually or in combination [152,153]. They can be manufactured via the curing of a mixture of polythiol, isocyanate, and organotin (Lewis acid) catalyst under a programmed temperature increase [154,155]. The reaction can be activated by a basic or a Lewis acid catalyst, as shown in Figure 15 [156]. Depending on the characteristic of the catalyst used, the activation occurs in two ways. If a basic catalyst is used, it activates the thiol by forming a nucleophilic activated anion, a species that attacks the isocyanate carbon [157]. If a Lewis acid is used as a catalyst, the coordination occurs with an isolated pair of oxygen or nitrogen (electrophilic activation) of the isocyanate that can be attacked by less nucleophilic substances, such as neutral thiols [154,158]. In general, thiols are more acidic than alcohols, and basic catalysts help form thiolate anions, which demonstrate higher nucleophilic properties than alkoxides. That is, the reactivity of thiols is higher than that of alcohols, so the thiol attack on isocyanates is much faster. However, a basic catalyst is not used in the manufacturing of highly refractive lenses because of its rapid reactivity, bubble generation, and lens appearance problems due to lack of uniformity of the cured specimen. In general, an organotin (Lewis acid) catalyst leading to a mild reaction is used in industrial highly refractive lens manufacturing.

### 6.3. Area of Epoxy Resin Hardener

Ciba Ltd. introduced a new class of thermosetting materials, “epoxy resins,” at the Swiss Industrial Fair in 1946. After approximately 75 years, epoxy resins are being applied in a wide range of industrial and scientific fields, from aerospace to general industrial adhesives. Epoxy has a three-membered ring structure, variously known as an epoxy, epoxide, or oxirane functional group, comprising two carbon atoms and one oxygen atom. Through the ring-opening reaction that causes crosslinking, the epoxy group is solidified (or cured), and the epoxy resin is classified as a thermosetting resin owing to the curing enabled by this chemical reaction. The composition of epoxy resin comprises at least two components: an epoxy resin and a hardener that causes curing. Fillers, dyes, solvents, diluents, plasticizers, curing accelerators, thermoplastic resins, and rubbers can also be added to improve the processing and properties of the cured resin [159].

Epoxy resins are the most widely applied resins among thermosetting polymers. Epoxy resins can react with various hardeners to form cured products with high strength and adhesion [160]. The nucleophilic substitution reaction is the most common method used for curing epoxy resins, and the hardener used may involve an active hydrogen compound in this reaction. Examples of active hydrogen compounds include amines, amides, thiols, phenols, etc. The curing reaction between the hardener and the epoxy resin proceeds via the opening of the oxirane ring in the epoxy resin [161,162,163,164,165].

Sharpless et al. confirmed the so-called spring-loaded properties of modified heterocyclic electrophiles for nucleophilic ring-opening reactions in several reactions [68]. One of these reactions, which occurs between thiol and epoxide, forms β-hydroxy thioether. This reaction process is known to be simple, modular, and regioselective, ensuring high yield. Therefore, the low molecular weight mixture of thiol and epoxide can be cured to produce a crosslinked thin film or a cured product in a bulk state [165,166,167].

Recently, it has been used to prepare a well-defined polymer with a desired structure or introduce a functional group for modification after polymerization. In some cases, the secondary alcohol produced is less reactive than the primary alcohol, but it is still available for the reaction, and its modification to produce polyfunctional compounds is being investigated. The chemical reaction mechanism of the polythiol–epoxy reaction is yet to be examined. The ring modification of the epoxide functional group makes the molecule vulnerable to nucleophilic attack to restore the ideal tetrahedral angle at every atom. Among various nucleophiles, polythiol must be converted into a thiolate anion to exist as a nucleophile that can participate in a ring-opening reaction. This is possible using a suitable base that can effectively deprotonate the thiol group of the polythiol. For example, hydroxide anions can deprotonate the thiol groups of polythiols. There is no competition between hydroxide and thiol because thiols are more acidic than alcohols considering an attack on the epoxide ring (the *p*Ka value of thiol is 5–10, which is lower than the *p*Ka value of alcohol, 15.7) [168,169]. Therefore, a rapid proton transfer from sulfur to oxygen occurs, and formed thiolate attacks the side with relatively little steric hindrance in the epoxide unit. Thus, an alkoxide unit is formed, and its stabilization to a secondary alcohol is shown in Figure 16. The reaction medium is typically moistened with water or has a protic nature in the curing system. Because thiol molecules are acidic, alkoxide units are protonated to high basicity (*p*Ka = 17) [170].

Epoxy adhesives [171] have broad applications in automotive [172,173,174], electronics [175,176], aircraft, [177,178,179], and other industries [180], and they require strength, such as the adhesion property obtained using other bonding methods, such as welding. In addition to adhesive strength, epoxy adhesives are cured quickly (within minutes at room temperature) and are preferred to be in a single-package configuration; therefore, they must have storage stability for at least several months [181]. In general, the main components of an epoxy adhesive are epoxy resin and the hardener, as for other epoxy-cured products [182]. The hardener can cure the epoxy resin under curing conditions. If the hardener reacts with the epoxy resin at room temperature, the epoxy resin and the hardener must be stored separately (two-component adhesive). Alternatively, if the curing agent does not react with the epoxy resin at a sufficient rate at room temperature without deformation, it can be stored in one container (one-component adhesive) [183].

Owing to the unique reactivity of polythiol and epoxy resin, polythiol hardeners are commercially available [184,185,186,187,188,189,190]. Polythiol hardeners can provide high reactivity at low temperatures and excellent adhesion to various materials, such as wood, fiberglass, ordinary glass, polymers, cement, ceramics, and metal. Polythiol and epoxy adhesives are commonly used as two-component adhesives owing to their ease of use and fast repair and tooling [191].

## 7. Conclusions and Remarks

In this review, an overview of the synthesis methods of polythiols and their industrial applications is provided, as highly versatile thiol chemistry allows various functionalization strategies, including thermal, photochemical, and redox processes. Although a glimpse into the various synthetic schemes that can be utilized in industrial applications is provided, the literature discussed in this review is by no means complete. However, we believe that the versatility of polythiols highlighted in this review may facilitate their implementation in materials science. Although the thiol–isocyanate and thiol–epoxy reaction described in this review merely represents the tip of the iceberg of the versatility of thiol-based click reactions, the guidance covered in this review may hold great promise for the expansion of their industrial applications. Finally, there is no doubt that further interesting developments in industrial applications of thiol-based click reactions will become the focus of research in this area.

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
