# Peer review of "Chemistry of Polythiols and Their Industrial Applications"

_materials, 2024, doi:10.3390/ma17061343_

Round 1

Reviewer 1 Report

Comments and Suggestions for Authors

After reading the manuscript, I believe that a more appropriate place for its publication should be Phosphours, sulfur and Silicon and the Related Elements of Journal of Sulfur Chemistry. If you decide to publish this manuscript in Polymers, the authors should supplement the reference list with many more titles from the last 5-7 years.  In addition, in the part devoted to synthesis methods, literature references should be extended by adding  monographic chapters discussing methods of synthesis of this class of organosulfur compounds ( for example chapters in Science of Synthesis and Comprehensive Organic Synthesis). Moreover, the authors should again prepare the revised manuscript very carefully to avoid substantive errors, which can be seen, for example, in Scheme 16 (thiolate anion can not be presented as RSH)

Reviewer 2 Report

Comments and Suggestions for Authors

The manuscript is in tne scope of the Journal. I recommended to accept after a revision. Follows are specific comments.

 1.     Section 1 “History of sulfur compounds” are not necessary to this review. At least, it should be shortened.

2.     Section 2 “Comparison of the physical properties of thiols and alcohols” is also too long. A brief comparison, such as bond energy, nucleophilicity and pKa is enough.

3.     Please check if Scheme 10b is correct.

4.     Figure 3 makes nonsense, please deleted.

5.     Thiol-yne click reaction should be introduced.

Reviewer 3 Report

Comments and Suggestions for Authors

The present review is dealing with the synthesis of polythiols and their reactivity (thiol-ene radical coupling, thia-Michael addition, epoxide opening, etc.) towards industrial applications: UV curing of polymers, high refractive index polymers for optical lenses, epoxy resins for adhesives. The manuscript is informative and quotes a large number of articles and books (177 refs). Moreover, it is well-written, although it appears quite didactic in the first sections (i.e. 1. History of sulfur compounds, 2. Comparison of the physical properties of thiols and alcohols, 3. Synthesis of thiol compounds). In other Journals, this manuscript will be considered a Tutorial Review.

In my opinion the paper can be accepted for publication in Polymers after the minor revisions listed below:

- a 2014 review dealing with polythiols should be quoted: "Polythiol copolymers with precise architectures: a platform for functional materials" Polymer Chemistry 2014, 5, 4601.

- line 54: the sentence "The oxygen atom of alcohol substituted with sulfur is called a thiol" is incorrect, thiol is a functional group or a molecule, not an atom.

- line 55: what does it mean "A thiol functional group is a thiol bonded to a carbon atom"?

- line 61: oxygen and sulfur are in the same group, not period, of the periodic table.

- Figure 3 showing simple glass and plastic lenses is useless and should be deleted.

- lines 413-421 and Figure 4, describing the manufacture of plastic lenses, is out of the scope of the review and should be deleted.

- Scheme 15: in the mechanism of the Lewis acid-catalysed addition of thiols to isocyanates the intermediate resulting from the addition of the Lewis acid must carry a positive, not a negative charge. The electrophilic intermediate will then react with the nucleophilic thiol to give first a sulfonium ion and finally the thiourethane as drawn.

- Figure 5 showing samples of a transparent epoxy resin is useless and should be deleted.

Round 2

Reviewer 2 Report

Comments and Suggestions for Authors

The manuscript has been well revised and I recommended to accept it at present form.

Author Response

We appreciate your valuable review for our manuscript.